# Virulence Factors and Antimicrobial Resistance in *Salmonella* Species Isolated from Retail Beef in Selected KwaZulu-Natal Municipality Areas, South Africa

Serisha Naidoo [1], Patrick Butaye [2,3], Tsolanku S. Maliehe [1], Kudakwashe Magwedere [4], Albert K. Basson [1] and Evelyn Madoroba [1,*]

[1] Department of Biochemistry and Microbiology, University of Zululand, Private Bag X1001, KwaDlangezwa 3886, South Africa; serishanaidoo25@gmail.com (S.N.); MalieheT@unizulu.ac.za (T.S.M.); BassonA@unizulu.ac.za (A.K.B.)

[2] Department of Pathobiology, Pharmacology and Zoological Medicine, Faculty of Veterinary Medicine, Ghent University, Salisburylaan 133, B9820 Merelbeke, Belgium; Patrick.Butaye@Ugent.be

[3] Department of Biomedical Sciences, Ross University School of Veterinary Medicine, Basseterre 00334, Saint Kitts and Nevis

[4] Directorate of Veterinary Public Health, Department of Agriculture, Land Reform and Rural Development, Pretoria 0001, South Africa; KudakwasheM@dalrrd.gov.za

[*] Correspondence: MadorobaE@unizulu.ac.za

**Abstract:** Salmonellosis and antimicrobial resistance caused by non-typhoidal *Salmonella* are public health concerns. This study aimed at determining prevalence, serovars, virulence factors and antimicrobial resistance of *Salmonella* from beef products. Four-hundred beef samples from 25 retail outlets in KwaZulu-Natal, South Africa were analyzed for *Salmonella* using standard methods, confirmation with matrix-assisted laser desorption ionization–time of flight and serotyping according to the White–Kauffmann–Le Minor scheme. The Kirby Bauer disk diffusion method was used to determine antimicrobial resistance against Cefotaxime, Kanamycin, Ampicillin, Amoxicillin, Trimethoprim Sulfamethoxazole, Ciprofloxacin, Chloramphenicol, Gentamicin Cefoxitin and Tetracycline. A polymerase chain reaction was performed to detect *invA*, *agfA*, *lpfA*, *hilA*, *sivH*, *sefA*, *sopE*, and *spvC* virulence genes. *Salmonella* was observed in 1.25% (5/400) of the samples. Four serovars (Enteritidis, Hadar, Heidelberg, Stanley) were identified. Almost all *Salmonella* were susceptible to all antimicrobials except *S.* Enteritidis isolate that was resistant to Tetracycline, Ampicillin and Amoxicillin. All *Salmonella* isolates carried at least two virulence factors. The findings indicate low *Salmonella* prevalence in meat from selected KZN retail beef; however, routine surveillance to monitor risk associated with virulence factors is required to mitigate potential outbreaks. The resistant *S.* Enteritidis highlights a need to routinely monitor antimicrobial resistance in order to enhance human health.

**Keywords:** *Salmonella enterica* serovars; virulence factors; antimicrobial resistance; beef and beef products; prevalence; food safety

## 1. Introduction

Animal-sourced foods contribute to a significant portion of the global foodborne-disease burden [1]. A higher infection rate with *Salmonella enterica* subspecies *enterica* serotypes in low to middle income countries has been reported possibly due to poor hygiene standards in food production as well as high rates of Human Immunodeficiency Virus (HIV), malaria, malnutrition and other clinical associations [2]. Among the various types of *Salmonella* spp., non-typhoid *Salmonella* (NTS) causes an estimated 80 million illnesses and 30,000 deaths each year [2,3].

The predominant cause of NTS infection in humans is animal-derived foods such as meat, eggs and milk [4]. *Salmonella* spp. commonly inhabit the gastrointestinal tract of healthy animals, mainly in the terminal ileum of the intestines. The gastrointestinal contents

may cross-contaminate the carcass surface during slaughter, mostly during evisceration and hide removal [5]. Although *Salmonella* contamination is more frequent in poultry and pork meat, beef has also been closely associated with outbreaks of salmonellosis over the years [5]. Meat handling, processing, transport, storage, distribution and preparation for consumption lead to higher levels of contamination [6]. Contamination from unsterile equipment, utensils and workers' hands as well as cross-contamination between different carcasses and meat types at abattoirs and retail outlets increases the prevalence of *Salmonella* in meat products [7].

Complexity has been reported regarding the virulence of NTS, as several genes involved in invasion and adhesion such as *agfA*, *lpfA*, *hilA*, *sivH* and *invA* have been detected [8]. The virulence involves an extensive set of factors, which are mainly encoded in virulence genes within *Salmonella* pathogenicity islands (SPIs) [9]. Twenty-four SPIs have been identified and are either located on the chromosomes or on plasmids, though not all virulence genes are embedded in these pathogenicity islands [9,10]. Furthermore, SPIs are capable of being horizontally transferred to other enteric bacteria, which may cause non-pathogenic microorganisms to become pathogenic [11]. Encoded virulence genes have various roles in pathogenesis, such as facilitating the attachment, invasion and intracellular survival of *Salmonella* and bypassing the hosts' defense mechanisms, resulting in diverse clinical symptoms based on the virulence genes present [9].

Globally, the extensive use of antimicrobial agents in animal production and clinical practices have contributed to the increasing pervasion of resistance of *Salmonella* strains toward antimicrobials, and thus the emergence of multi-drug resistant (MDR) strains [12]. Outbreaks caused by MDR- NTS have been linked to higher rates of hospitalization and treatment failures of patients and thus higher morbidity and mortality rates over the years [13,14].

In South Africa, the hygiene, quality of meat and number of units that can be slaughtered are primarily regulated via the Meat Safety Act, 2000 (Act No. 40 of 2000) and the Foodstuffs, Cosmetics and Disinfectants Act, 1972 (Act No. 54 of 1972). The KwaZulu-Natal province was selected as it is the second most important cattle producer in South Africa and a significant contributor to the supply of beef in the country [15]. With the predominant source of *Salmonella* infection being animal-originating foods [14] and South Africa having an ever-expanding beef industry [16], the raw beef production chain, from farm to abattoirs and retail outlets should be regularly investigated. Detailed knowledge of the virulence factors of *Salmonella* serovars from different beef products in KZN retail is limited [17]. The aim of this study was to expand the knowledge on the prevalence, serovars, antimicrobial resistance and virulence factors from *Salmonella* isolates that were recovered from beef and beef products among selected retail outlets from the KwaZulu-Natal province in South Africa. The samples represented various degrees of processing, including raw intact beef, raw beef organs, raw processed beef and ready to eat beef products such as biltong and cold meats. The information provides knowledge about the pathogenicity of *Salmonella* species that circulate in the food value chain and could pose a potential health threat to consumers.

## 2. Materials and Methods

### 2.1. Ethical Approval

Ethical approval for this study was obtained from the University of Zululand with approval reference number UZREC171110-030 PGM 2019/109.

### 2.2. Study Design

A cross-sectional study was undertaken in four local municipalities in King Cetshwayo district and three local municipalities within iLembe district in KwaZulu-Natal (KZN) province, South Africa (Figure 1). A total of 400 samples were obtained from 25 facilities (retail stores and butcheries) of different sizes across King Cetshwayo and Ilembe between October 2019 and December 2020. The samples included raw intact beef cuts (*n* = 53), raw beef organ meats such as kidneys, intestines, tripe, liver, lung, and spleen (*n* = 169), raw processed beef products (*n* = 110) and ready-to-eat beef products (*n* = 68). All sample

sources were coded in order to maintain strict confidentiality. The samples were packed in insulated boxes on ice, followed by transportation to the Department of Biochemistry and Microbiology at University of Zululand.

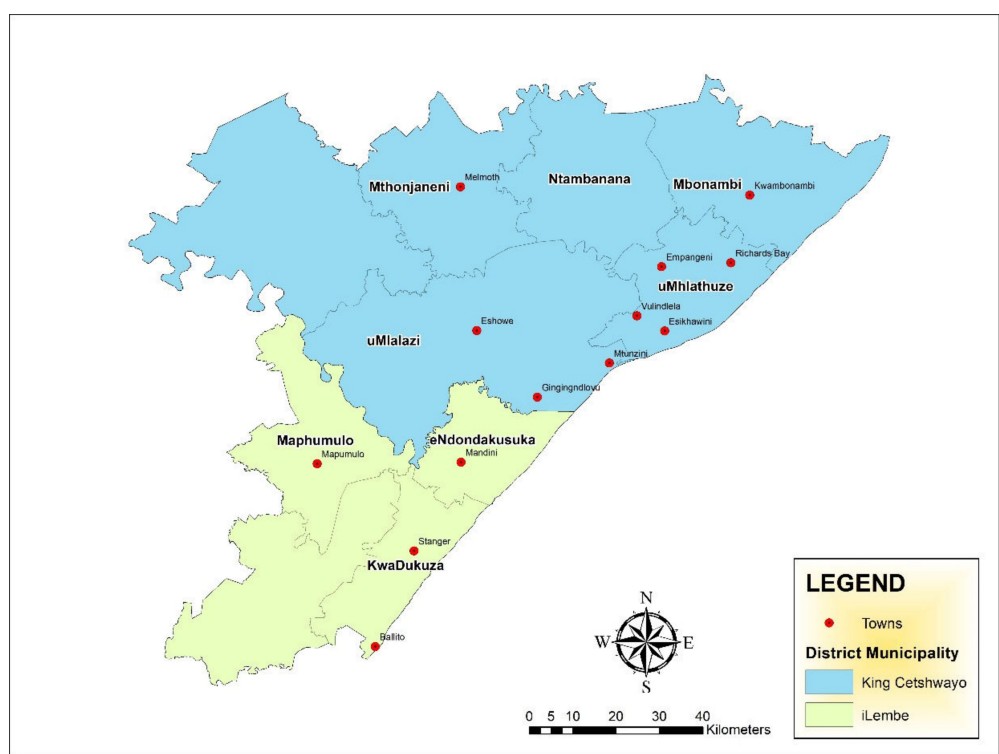

**Figure 1.** Map of King Cetshwayo district and iLembe district in KZN province, South Africa showing the towns were sample were obtained (Source: Naidoo, unpublished).

*2.3. Sample Size Determination*

The sample size was determined using an equation for cross-sectional studies [18,19]:

$$n = \frac{Z_{(1-\frac{\alpha}{2})}^2 p(1-p)}{d^2} \tag{1}$$

where

$n$ = sample size, $Z_{(1-\frac{\alpha}{2})}$ = standard normal variate (at 5% type 1 error, yielding 1.96);

$p$ = the expected proportion in the population based on previous studies and;

$d$ = absolute error.

There was no official record for the expected proportion of *Salmonella* in South African beef samples from retail outlets, hence an estimated prevalence of 50% was used. The sample size was calculated with an absolute error of 5% to allow for a slight variation in samples.

$$n = \frac{1.96^2 \times 0.5(1-0.5)}{0.05^2} \tag{2}$$

$n = 384.16 \approx 384$.

Using these values, the estimated minimum sample size required was calculated to be 384. A sample size of 400 was used in this study.

*2.4. Microbiological Analysis*

2.4.1. Isolation and Identification of Salmonella

The detection, isolation and identification of *Salmonella* spp. was based on the International Organization for Standardization (ISO) 6579-1:2017 [20]. For pre-enrichment, each

sample was weighed to obtain a mass of 25 g and homogenized in 225 mL buffered peptone water, followed by incubation for 18 h $\pm$ 2 h at 37 $°$C $\pm$ 2 $°$C. For selective enrichment, 1 mL and 0.1 mL of pre-enriched broths were inoculated into 10 mL Muller Kauffmann tetrathionate novobiocin (MKTTn) broth and Rappaport-Vassiliadis Soya (RVS) broths, respectively. The inoculated MKTTn and RVS broths were incubated for 24 h $\pm$ 2 at 37 $°$C $\pm$ 2 $°$C and 41.5 $°$C $\pm$ 2 $°$C, respectively. The broths were streaked onto Brilliant Green Agar (BGA) and Xylose Lysine Deoxycholate (XLD) agar, followed by incubation at 37 $°$C $\pm$ 2 $°$C for 24 h $\pm$ 2. Presumptive *Salmonella* colonies on XLD plates were identified as having a bright pink colour with/without a black center. Presumptive *Salmonella* colonies appeared bright pink on BGA plates. When available, up to four isolated colonies were picked per suspected positive sample. All presumptive *Salmonella* were preserved and stored in sterile Eppendorf tubes containing sterile nutrient broth supplemented with glycerol to a final concentration of 30% until required for further analysis.

### 2.4.2. Salmonella Identification Using MALDI-TOF MS and VITEK MS

Presumptive *Salmonella* isolates were identified using MALDI-TOF MS technology [21] and VITEK technology.

### *2.5. Serotyping*

*Salmonella* isolates that were confirmed using MALDI-TOF MS and VITEK technology were sub-cultured and serotyped according to the Kauffman-White-Le Minor classification scheme [22] as previously described [23]. Briefly, *Salmonella* colonies were suspended in sterile physiological saline and evaluated for auto-agglutination prior to serotyping using 'O' and 'H' antisera.

### *2.6. Antimicrobial Susceptibility Testing*

The antimicrobial susceptibility patterns of each *Salmonella* isolate were determined using the Kirby-Bauer disk diffusion method [24]. Briefly, the optical density (OD) of bacterial suspensions at the exponential growth stage were determined at 625 nm using spectrophotometer (Merck Spectroquant Pharo, Darmstadt, Germany) and adjusted to 0.08–0.13. The suspensions were compared to 0.5 McFarland turbidity standard. The standardized *Salmonella* suspensions were inoculated onto Mueller Hinton (MH) agar using sterile cotton swabs. The plates were left to stand at room temperature for 5 min to allow them to dry prior to the insertion of antimicrobial disks. The ten antimicrobial disks (Table 1) were placed onto the MH plates, followed by incubation according to the time and temperature recommended by the CLSI guidelines for each antimicrobial [25]. The inhibition zones were measured using calipers and recorded as either susceptible, intermediate or resistant to each drug based on the Clinical Laboratory Standard guidelines [25].

**Table 1.** Disk content of antimicrobials.

| Antimicrobial Agent | Abbreviation | Disk Content (µg) |
|---|---|---|
| Amoxicillin | A | 25 |
| Ampicillin | AP | 10 |
| Cefoxitin | FOX | 30 |
| Cefotaxime | CTX | 30 |
| Chloramphenicol | C | 30 |
| Ciprofloxacin | CIP | 5 |
| Gentamicin | GM | 10 |
| Kanamycin | K | 30 |
| Tetracycline | TET | 30 |
| Trimethoprim sulfamethoxazole | TS | 25 |

*2.7. Polymerase Chain Reaction (PCR) for Detection of Virulence Genes*

2.7.1. DNA Extraction

The Quick-DNA™ Miniprep Kit was used for the rapid extraction of DNA from the confirmed *Salmonella* isolates in accordance with the manufacturer's instructions.

2.7.2. PCR Analysis

The *Salmonella* isolates were subjected to Polymerase Chain Reaction (PCR) to screen for the presence of the following eight virulence genes: *invA*, *agfA*, *IpfA*, *hilA*, *sivH*, *sefA*, *sopE*, and *spvC* as previously described [8] with slight modifications. Forward and reverse primer sequences for each gene are shown in Table 2.

**Table 2.** Primers used to screen for each of eight virulence genes in *Salmonella* spp. Using PCR *.

| Target Gene | Sequence | Amplicon Size (bp) |
|:---:|:---:|:---:|
| *invA* | F-GTGAAATTATCGCCACGTTCGGGCAA R-TCATCGCACCGTCAAAGGAACC | 284 |
| *agfA* | F-TCCACAATGGGGCGGCGGCG R-CCTGACGCACCATTACGCTG | 350 |
| *lpf*A | F-CTTTCGCTGCTGAATCTGGT R-CAGTGTTAACAGAAACCAGT | 250 |
| *hilA* | F-CTGCCGCAGTGTTAAGGATA R-CTGTCGCCTTAATCGCATGT | 497 |
| *sivH* | F-GTATGCGAACAAGCGTAACAC R-CAGAATGCGAATCCTTCGCAC | 763 |
| *sef*A | F-GATACTGCTGAACGTAGAAGG R-GCGTAAATCAGCATCTGCAGTAGC | 488 |
| *sop*E | F-GGATGCCTTCTGATGTTGACTGG R-ACACACTTTCACCGAGGAAGCG | 398 |
| *spvC* | F-CCCAAACCCATACTTACTCTG R-CGGAAATACCATCTACAAATA | 669 |

* References: [8,26–32].

The 20 μL PCR master mix consisted of 10 μL NEB OneTaq 2X MasterMix with Standard Buffer (Catalogue No. M0482S), 1 μL Genomic DNA (10–30 ng/μL), 7 μL Nuclease free water (Catalogue No. E476) and 1 μL of each primer (10 μM forward and 10 μM reverse). The mixtures were subjected to amplification in a thermocycler machine and underwent conditions consisting of an initial denaturation at 94 °C for 5 min, followed by 35 cycles each of denaturation at 94 °C for 30 s, annealing at 50 °C for 30 s, elongation at 68 °C for 1 min and a final extension at 68 °C for 10 min. The amplicons were then held at 4 °C.

2.7.3. Agarose Gel Electrophoresis

The PCR amplicons were subjected to agarose gel electrophoresis using 1.5% ethidium bromide stained agarose gel at 3 V/cm for approximately 1 h. A100 bp DNA ladder was used to estimate the size of the amplicons. Gels were viewed and documented under ultraviolet light using a gel documentation system.

*2.8. Reference Strains*

The following reference strains were included for all experiments: *S.* Typhimurium ATCC® 13076™, *E. coli* ATCC® 25922™ and field strains that are positive or negative for the required analysis.

*2.9. Statistical Analysis*

Confidence intervals for the population proportions, with a lower and higher bound, were calculated using the exact binomial distribution at a confidence level of 95% using Excel.

## 3. Results

On average, the prevalence of *Salmonella* spp. in beef and beef products was 1.25% (*n* = 5/400; Confidence interval (CI) = 0.4–3). The samples were mainly of gastrointestinal origin (3/400; 0.75%; CI = 0.2–2; Table 3), followed by beef kidney (*n* = 1/400; 0. 25%; CI = 0–1) and beef rib (*n* = 1/400; 0.25%; CI = 0–1). From these five *Salmonella* positive samples, five isolates were obtained. The isolates belong to four different serovars, namely *S*. Stanley (*n* = 2), *S*. Heidelberg (*n* = 1), *S*. Hadar (*n* = 1) and *S*. Enteritidis (*n* = 1) (Table 3).

**Table 3.** *Salmonella enterica* subspecies *enterica* serovars from beef products.

| Sample Type | Serovar |
| --- | --- |
| Beef intestines | Stanley |
| Beef tripe | Heidelberg |
| Beef tripe | Hadar |
| Beef Kidney | Enteritidis |
| Beef Rib | Stanley |

Drug resistance was observed for *S*. Enteritidis only. The *S*. Enteritidis was resistant to tetracycline and the aminopenicillins, which are ampicillin and amoxicillin. The *S*. Stanley, *S*. Heidelberg, and *S*. Hadar were susceptible to ampicillin, amoxicillin, chloramphenicol, cefotaxime, gentamicin, ciprofloxacin, kanamycin, tetracycline, cefoxitin, and trimethoprim—sulfamethoxazole.

*Salmonella* isolates tested positive for a minimum of two virulence genes and some serovars tested positive for up to six out of eight virulence factors (Table 4). The virulence gene profile of each of the isolates was unique, without a repetition of the same combination of virulence genes (Table 4). Virulence gene *agfA* was predominant, and was observed in all of the five isolates. Both the *invA* and *hilA* genes were found in all except one isolate (Table 4). One of the two *S*. Stanley strains contained six of the eight virulence genes, whilst *S*. Heidelberg harbored the least (2/8) number of virulence genes. The *spvC* gene was not detected in any of the six *Salmonella* isolates in this study.

**Table 4.** PCR results for eight virulence genes from six *Salmonella* isolates obtained from beef products.

| Sample Type | Serovar | Virulence Gene | | | | | | | |
| --- | --- | --- | --- | --- | --- | --- | --- | --- | --- |
| | | *invA* | *AgfA* | *LpfA* | *HilA* | *sivH* | *SefA* | *sopE* | *spvC* |
| Beef Intestines | *S*. Stanley | + | + | − | + | + | − | − | − |
| Beef tripe | *S*. Heidelberg | − | + | − | + | − | − | − | − |
| Beef intestines | *S*. Hadar | + | + | + | + | + | − | − | − |
| Beef kidney | *S*. Enteritidis | + | + | + | − | − | + | − | − |
| Beef rib | *S*. Stanley | + | + | + | + | − | + | + | − |

+: Refers to positive PCR test for the target virulence gene; −: Refers to negative PCR test for the target virulence gene.

*Detection of Virulence Genes Using PCR*

Figures 2–4 show examples of agarose gel pictures showing PCR amplicons for *invA*, *hilA* and *sivH* genes, respectively.

Lanes: 1: 50 bp DNA ladder; 2; 3 and 5 show PCR amplicons of samples positive for *invA* gene. Lane 4 shows a negative result for *invA* (*S*. Heidelberg). The serovars in lanes 2,3 were *S*. Stanley and lane 5 contained *S*. Hadar. The *invA* gene is 284 bp in length.

Lane 1 contained the 50 bp DNA ladder and the remainder of the lanes had PCR amplicons run. Lanes 2 (*S*. Stanley) and 5 (*S*. Hadar) were positive for the 763 bp *sivH* gene.

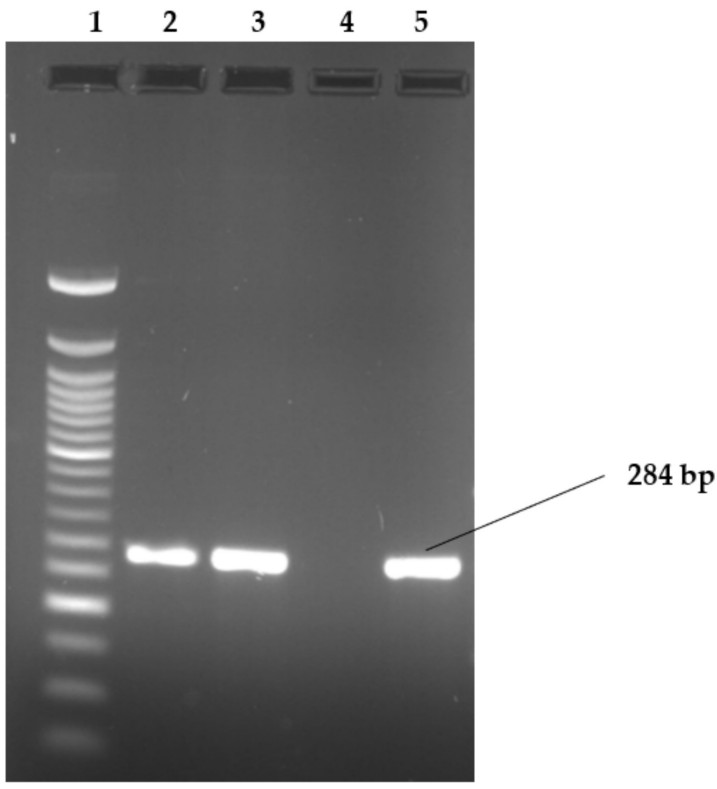

**Figure 2.** Image showing examples of *invA* gene amplicons observed on agarose gel.

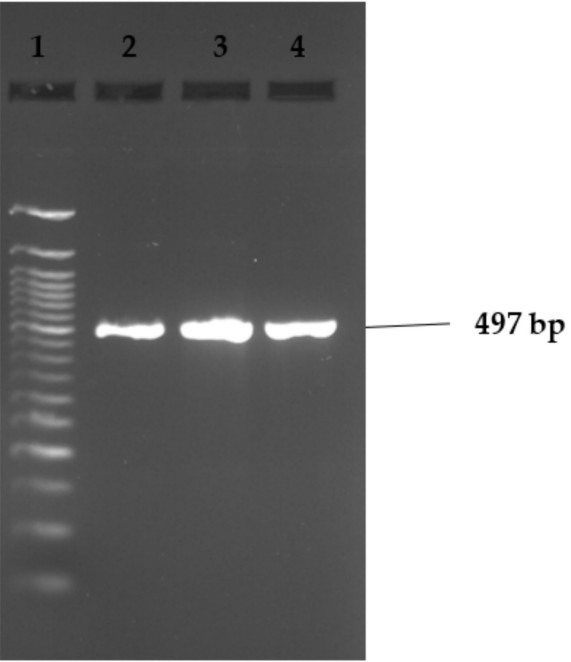

**Figure 3.** Image showing example of *hilA* gene amplicons observed on agarose gel. Lane 1 contains the 50 bp DNA ladder and lanes 2–4 show positive results for the *hilA* gene (497 bp). Lanes 2 and 3 contained amplicons for *S.* Stanley isolates, while lane 4 contained *S.* Hadar.

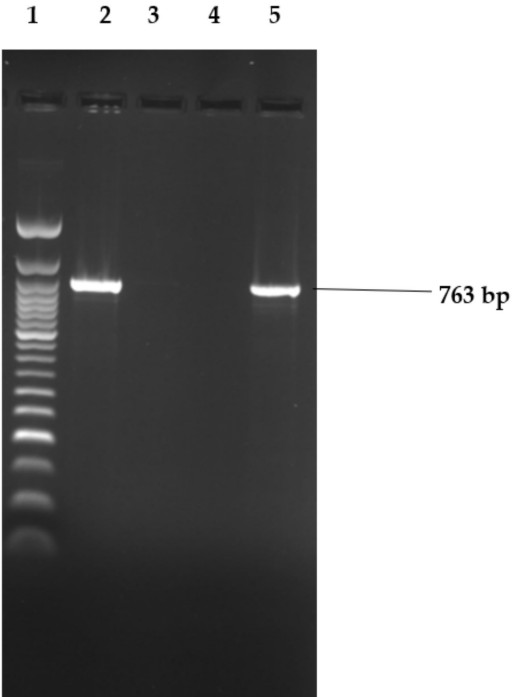

**Figure 4.** Image showing electrophoresis results for *sivH* gene amplicons.

### 4. Discussion

The overall low prevalence of *Salmonella* (1.25%) in this study was similar to other studies in in KwaZulu-Natal [33], where abattoirs and meat from different animal species were also involved, but no offal. In this study, the majority of the *Salmonella* originated from offal, and thus the former study may probably somehow show an underrepresentation of the prevalence on beef products. The total prevalence found in the former study was 3.2%, with a lower prevalence for beef meats in KwaZulu-Natal and in the Western Cape [33]. In other African countries such as Botswana, a higher prevalence was found, though these studies are older and the epidemiological situation might have changed [34]. The low prevalence found in this study may be due to the fact that samples were not obtained from informal butcheries. A study carried out in Senegal found *Salmonella* contamination in 63% of the total beef samples tested, 43% (101/236) in slaughterhouses and 87% (174/199) in retail outlets, with 97% of the tested retail outlets having at least one positive sample. The study identified reasons for the high contamination levels as incorrect storage temperatures, poor hygiene of workers and equipment and unwashed cutting tables which could have acted as reservoirs for *Salmonella* to spread to other equipment through flies or direct contact [35]. The presence of *Salmonella* spp. in retail meat samples, even in low amounts, creates the basis for potential spread. Regular surveillance should be carried out in both the formal and informal beef sector in South Africa to ensure conditions do not lead to outbreaks.

This is the first study on beef organ meats in KwaZulu-Natal, and they have been proven the most contaminated, probably due to intestinal contamination and their association with the gut. However, tripe and intestines are usually thoroughly cooked, hence the chances of *Salmonella* survival are reduced. Even so, tripe and intestines may cross contaminate different surfaces and utensils, which may result in infection. A higher prevalence in such organ meat compared to other beef cuts has been demonstrated before in Burkina Faso and Botswana [34,36]. Ready-to-eat meat samples, which included sample types such as cold meats, biltong (dried, cured meat) and smoked meats did not contain *Salmonella*, which is not surprising, as these products undergo further processing.

The rib-meat sample that tested positive for *Salmonella* in this study could have been contaminated during one of the many stages of beef processing due to the complex nature of meat contamination [23]. It is likely that the contamination of rib pieces occurred during cutting and packaging, as large quantities may have been cut using the same equipment and on the same surfaces before being placed into their individual packages.

Surprisingly, the current investigation did not yield *Salmonella* from raw processed products. Sausages are generally produced in bulk and can contain ground meat of various different cuts of beef; hence our findings may indicate that strict hygiene processes were implemented during the production process in the selected municipalities and the cut used were not contaminated.

From the five *Salmonella* isolates, four different serovars were identified. *S.* Stanley (*n* = 2) was the most prevalent serotype and *S.* Hadar, *S.* Heidelberg and *S.* Enteritidis, were represented by a single isolate. A recent review on serotypes in different African countries found that the most isolated serovars from meat and meat products, including beef were *S.* Typhimurium and *S.* Enteritidis [37]. In a previous South African study undertaken between 2011 and 2012, the most common *Salmonella* serovar found on cattle carcasses and hides was *S.* Enteritidis [23]. While *S.* Heidelberg was also found [23], *S.* Stanley and *S.* Hadar were rather uncommon in Africa [37].

All *Salmonella* isolates were susceptible to cefoxitin, cefotaxime, chloramphenicol, ciprofloxacin, gentamicin, kanamycin and trimethoprim-sulfamethoxazole. However, it is important to note that first and second generation cephalosporins (cefoxitin and cefotaxime) and aminoglycosides (gentamicin and kanamycin) that appear effective in vitro, may be ineffective in clinical treatment and thus cannot be reported as susceptible. Widespread resistance against tetracyclines may arise as a result of selection pressure caused by the use of these agents in clinical and veterinary treatment, prophylaxis and as growth promoters for livestock [38]. Previous studies in South Africa found a high level of resistance to tetracyclines and ampicillin in *Salmonella* from cattle meats [17,23]. Additionally, in other African countries, resistance against these antibiotics in *Salmonella* was highest [37]. However, the small sample of five *Salmonella* isolates from this study does not provide a good overview of the potential threat of multi-drug resistance in *Salmonella* in KZN.

Virulence genes are crucial for *Salmonella* pathogenesis and they manifest differently within the different hosts [39]. The five *Salmonella* isolates were screened for the presence of eight virulence genes, namely, *invA*, *agfA*, *lpfA*, *hilA*, *sivH*, *sef*A, *sop*E, and *spvC* using PCR. The *invA* gene, responsible for the invasion of epithelial cells at the onset of infection, is typically present in all *Salmonella* spp. and is therefore used as a target for rapid identification using PCR [8]. However, in the present study, the *invA* gene was not observed in one of the five isolates (*S.* Heidelberg). Although *invA*-negative *Salmonella* are considered rare, they have been identified in other studies [40,41]. Nevertheless, *invA*-negative strains may use other mechanisms for invasion [42,43].

The virulence gene that was common to all five isolates was *agfA*, which encodes fimbriae that increases the adhesion of the bacterium to intestinal cells and is also involved in biofilm formation [8]. Other genes were variably present and *spvC*, a plasmid mediated gene involved in vertical transmission [8] was not present in any of the six isolates. All isolates, including the three *S.* Stanley isolates, had a different virulence profile, indicating a different potential for infection of animals and humans [9].

The limitation of this study is that due to the low prevalence of *Salmonella*, we could not give a representative view on the characteristics of *Salmonella* on beef meat and beef organs. The serotypes detected in this study might not be representative of the serotypes that can be present. Antimicrobial resistance was low, with only one strain being resistant, though we only investigated five strains, which is not representative of the potential resistance that can be found in *Salmonella* from beef products. Similarly, the virulence genes detected in this study may be an underrepresentation of the virulence of *Salmonella* from beef.

## 5. Conclusions

In this study, we determined the prevalence, serovars, antimicrobial resistance profiles and selected virulence factors of *Salmonella* spp. in raw intact beef, raw processed been, ready-to-eat beef products and beef organs from selected retail outlets and butcheries in KZN, South Africa. Overall, a low prevalence (1.25%) of *Salmonella* spp. was observed and the isolated serovars were Enteritidis, Hadar, Heidelberg and Stanley. Beef organs had the highest degree of contamination, hence there is a need to be vigilant when processing offal. For this reason, it is important that regular surveillance be performed for food pathogens in meat products to ensure than prevalence remains low and the risk of possible future outbreaks is reduced. Drug resistance was observed in *S*. Enteritidis, which is one of the most common serovars associated with salmonellosis. The presence of virulence genes is crucial in *Salmonella* pathogenicity and it is important to have an understanding of the virulence profiles as this contributes to epidemiological knowledge about the potential severity of infections. Future studies should include a larger sample size by taking the informal sector into consideration.

**Author Contributions:** Conceptualization, E.M., P.B.; methodology, E.M., P.B. and S.N.; formal analysis, E.M., S.N. and P.B.; writing—original draft preparation, S.N.; writing—review and editing, E.M., P.B., K.M., A.K.B., T.S.M.; supervision, E.M., P.B., A.K.B. and T.S.M.; funding acquisition, E.M. All authors have read and agreed to the published version of the manuscript.

**Funding:** This study was funded by the Red Meat Research and Development South Africa (RM-RDSA) and the Department of Trade and Industry-Technology and Human Resource for Industry Programme (DTI-THRIP). The DTI-THRIP grant was awarded to Evelyn Madoroba through grant number THRIP/22/30/11/2017. We thank the relevant municipalities that were involved in the study.

**Institutional Review Board Statement:** Ethical approval for this study was obtained from the University of Zululand with approval reference number UZREC171110-030 PGM 2019/109.

**Informed Consent Statement:** Not applicable.

**Acknowledgments:** This study was funded by the Red Meat Research and Development South Africa (RMRD SA) and the Department of Trade and Industry-Technology and Human Resource for Industry Programme (DTI-THRIP). The DTI-THRIP grant was awarded to Evelyn Madoroba through grant number THRIP/22/30/11/2017. We thank the relevant municipalities that were involved in the study.

**Conflicts of Interest:** The authors declare no conflict of interest.

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
