# Peer review of "Virulence Factors and Antimicrobial Resistance in Salmonella Species Isolated from Retail Beef in Selected KwaZulu-Natal Municipality Areas, South Africa"

_applsci, doi:10.3390/app12062843_

Round 1

Reviewer 1 Report

The manuscript is a communication about prevalence, serovars, virulence factors and anti- 14 microbial resistance of Salmonella in beef products. The topic is consistent with the scope of the journal and the manuscript is generally well-written and structured. The research is detailed, material and methods are well performed and the conclusions are of interest. The manuscript is easy to understand by readers and it represents a field that need to be explored. The communication just need few changes and a little revision of the “Results” section organization. Specific suggestion are showed in the attached file.

Author Response

Thank you for another opportunity to revise our manuscript. We hereby submit the revised manuscript entitled “Virulence Factors and Antimicrobial Resistance in Salmonella species isolated from retail Beef in selected KwaZulu-Natal municipality areas, South Africa". Revisions have been made in accordance with comments that have been provided. We have provided point-by-point responses for each comment. We thank all the individuals who have contributed to the comments that will enhance the quality of our manuscript. 

Responses to Reviewers’ Comments:

Reviewer 1

We thank reviewer 1 for contributing to improving the quality of our manuscript.

Comment 1: Line 64: Please cite:

  • Festa, R., Ambrosio, R. L., Lamas, A., Gratino, L., Palmieri, G., Franco, C. M., ... & Anastasio, A. (2021). A Study on the Antimicrobial and Antibiofilm Peptide 1018-K6 as Potential Alternative to Antibiotics against Food-Pathogen Salmonella enterica. Foods10(6), 1372.

Response 1: The citation is included in the manuscript

Comment 2: Line 93: Do you refer to fig. 1?

Response 2: Yes, we refer to fig. 1. The typographical error has been corrected

Comment 3: Line 152: Please correct the style and add the paragraph.

Response 3: The correct style has been used

Comment 4: Line 260-320: The number of figures is not proportional to the text length. Please try to explain also in the text the results showed in the figures.

Response 4: Additional descriptions have been added. The revised text is as follows:

On average, the prevalence of Salmonella spp. in beef and beef products was 1.25% (n = 5/ 400; Confidence interval (CI) = 0.4 - 3). The samples were mainly of gastrointestinal origin (3/400; 0.75%; CI = 0.2 - 2; Table 3), followed by the same number for beef kidney (n = 1/400; 0. 25%; CI = 0 - 1) and beef rib (n = 1/400; 0.25%; CI = 0 -1). From these 5 Salmonella positive samples, six isolates were obtained. The 6 isolates belong to 4 different serovars, namely S. Stanley (n = 3), S. Heidelberg (n = 1), S. Hadar (n = 1) and S. Enteritidis (n = 1) (Table 3).

Drug resistance was observed for S. Enteritidis only. The S. Enteritidis was resistant to tetracycline and the aminopenicillins, which are ampicillin and amoxicillin. The S. Stanley, S. Heidelberg, and S. Hadar were susceptible to ampicillin, amoxicillin, chloramphenicol, cefotaxime, gentamicin, ciprofloxacin, kanamycin, tetracycline, cefoxitin, and trimethoprim – sulfamethoxazole.

Salmonella isolates tested positive to a minimum of 2 virulence genes and some serovars tested positive for up to 6 out of 8 virulence factors (Table 4). The virulence gene profile of each of the isolates was unique without repetition of the same combination of virulence genes (Table 4). Virulence gene agfA was predominant, and was observed in all of the 6 isolates. Both the invA and hilA genes were found in all except one isolate (Table 4). One of the three S. Stanley strains contained six of the 8 virulence genes, whilst S. Heidelberg harbored the least (2/8) number of virulence genes. The spvC gene was not detected in any of the 6 Salmonella isolates from this study.

Comment 5: Line 271: Please check the paragraph title position.

Response 5: The position of the title has been adjusted

Comment 6: Line 376: Please cite:

  • Peruzy, M. F., Capuano, F., Proroga, Y. T. R., Cristiano, D., Carullo, M. R., & Murru, N. (2020). Antimicrobial susceptibility testing for salmonella serovars isolated from food samples: Five-year monitoring (2015–2019). Antibiotics, 9(7), 365.

Response 6: The citation has been added

Sincerely,

Prof. Evelyn Madoroba

(PhD Microbiology; MSc Biotechnology; Masters in Business Leadership; PGDip.Higher Education)

Reviewer 2 Report

applsci-1582360

In this manuscript, a study aimed at determining prevalence, serovars, virulence factors and antibiotic resistance of Salmonella strains isolated from beef products, was carried out. Non-typhoids Salmonella strains in meat and organe meat samples collected from detail markets in from KwaZulu-Natal province in South Africa were detected. Investigations evidenced, albeit at low levels, the presence of non-typhoid strains of the genus Salmonella. The obtained results pointed out the presence of non-typhoid strains of Salmonella and for the first time, samples of organ meat were tested for the presence of the bacterial pahogen in this area of South Africa.

Considering that non-typhoid Salmonella strains are responsible for fever and infection in child before 2 years, the presence of this pathogen in these meat samples could represent a challenge for public health in low and medium income countries.

In the abstract sections, the authors refer to “… monitor antimicrobial resistance in order to enhance ‘One Health’”. The concept of One Health was not treated further along the manuscript, instead, describing more in details aspects related to One Health in preventing non-typhoid Salmonella infections, could add insights even to the methods proposed to face with challenges of the diffusion of these pahogenic bacterial strains.

The manuscript notes that the non-typhoid-type Salmonella contamination situation was first described on beef organ meats in KwaZulu Natal, South Africa, and proved to be the most contaminated.

The study reported that the presence of virulence genes must be considered as it contributes to epidemiological knowledge about potentiality of infections.

This Communication is of interest both as a local study and as a survey model to be used in other contexts.

Revisions

Lines 40 and 51: “non-typhoid Salmonella” change to “NTS”(?) as at line 38 the acronym NTS was indicated and at line 65 it ihas been used ‘MDR- NTS’;

Lines 104-167: in the Material and Methods section, from pharagraphers 2.3 to 2.5, the references are reported in extenso and not as numbers, please standardize along the text of the manuscript;

Lines 161-163: the ten antibiotics tested with the isolated strains were reported in Table1, it is pleonastic to introduce them also into the text of the manuscript;

line 215: change all the reported genes to Italic style;

line 217: “table 2” change to “Table 2”;

Lines 253, 254 and 264: change species names (S. and E.coli) to Italics;

lines 263 and 265: table 3 and table 4, change to Table 3 and Table 4;

Lines 349-350: “… which is not surprising since the different processes applied to these meant” please explain better this sentence.

Author Response

Thank you for another opportunity to revise our manuscript. We hereby submit the revised manuscript entitled “Virulence Factors and Antimicrobial Resistance in Salmonella species isolated from retail Beef in selected KwaZulu-Natal municipality areas, South Africa". Revisions have been made in accordance with comments that have been provided. We have provided point-by-point responses for each comment. We thank all the individuals who have contributed to the comments that will enhance the quality of our manuscript. 

Responses to Reviewers’ Comments:

Reviewer 2

 We thank reviewer 2 for contributing to improving the quality of our manuscript.

 Response to comment about the use of the term ‘One Health’: The term has been replaced with the words human health.

Comment 1: Lines 40 and 51: “non-typhoid Salmonella” change to “NTS”(?) as at line 38 the acronym NTS was indicated and at line 65 it ihas been used ‘MDR- NTS’;

Response 1: The change from “non-typhoid Salmonella” to “NTS” has been implemented.

Comment 2: Lines 104-167: in the Material and Methods section, from pharagraphers 2.3 to 2.5, the references are reported in extenso and not as numbers, please standardize along the text of the manuscript.

Response 2: The format of the references has been revised and numbers are included instead of names.

Comment 3: Lines 161-163: the ten antibiotics tested with the isolated strains were reported in Table1, it is pleonastic to introduce them also into the text of the manuscript;

Response 3: The names of the antibiotics have been removed from the text.

Comment 4: line 215: change all the reported genes to Italic style;

Response 4: The reported genes are written in Italics

Comment 5: line 217: “table 2” change to “Table 2”;

Response 5: Upper case ‘T’ is used for the word Table

Comment 6: Lines 253, 254 and 264: change species names (S. and E. coli) to Italics;

Response 6: The names are written in Italics

Comment 7: lines 263 and 265: table 3 and table 4, change to Table 3 and Table 4;

Response 7: Upper case ‘T’ is used for the word Table

Comment 8: Lines 349-350: “… which is not surprising since the different processes applied to these meant” please explain better this sentence.

Response 8: The statement has been revised to read as follows: ‘Ready-to-eat meat samples, which included sample types like cold meats, biltong (dried, cured meat) and smoked meats did not contain Salmonella, which is not surprising as these products undergo further processing.’

Sincerely,

Prof. Evelyn Madoroba

(PhD Microbiology; MSc Biotechnology; Masters in Business 

Reviewer 3 Report

What is missing from the research is statistical analysis. In order to be validated the authors should such analysis to assess the relationship between virulence genes of different serotypes. The authors didn't mention the percentage of different positive samples: how many tripe, kidney, and ribs samples were collected from the total 400? The authors are inconclusive when presenting the results in tables 3 and 4 because in table 4 they presented virulence genes of S. Stanley in beef intestines, but in table 4 is not;  the same regarding S Hadar in beef intestines, and in table 3, S. Hadar was detected in beef tripe.  Also, the results regarding antibiograms are not presented, there are some mentions in the discussion section. And for this also statistical analyses should be done for this.

Author Response

Thank you for another opportunity to revise our manuscript. We hereby submit the revised manuscript entitled “Virulence Factors and Antimicrobial Resistance in Salmonella species isolated from retail Beef in selected KwaZulu-Natal municipality areas, South Africa". Revisions have been made in accordance with comments that have been provided. We have provided point-by-point responses for each comment. We thank all the individuals who have contributed to the comments that will enhance the quality of our manuscript. 

Responses to Reviewers’ Comments:

Reviewer 3

We thank reviewer 3 for contributing to improving the quality of our manuscript.

Comments: What is missing from the research is statistical analysis. In order to be validated the authors should such analysis to assess the relationship between virulence genes of different serotypes. The authors didn't mention the percentage of different positive samples: how many tripe, kidney, and ribs samples were collected from the total 400? The authors are inconclusive when presenting the results in tables 3 and 4 because in table 4 they presented virulence genes of S. Stanley in beef intestines, but in table 4 is not; the same regarding S Hadar in beef intestines, and in table 3, S. Hadar was detected in beef tripe.  Also, the results regarding antibiograms are not presented, there are some mentions in the discussion section. And for this also statistical analyses should be done for this.

Response to Comments: Proportions of the different meat types (positive tripe, kidney, and rib samples out of the total 400) have been included in the text. Confidence intervals have been included. The source of the S. Hadar in Tables 3 and 4 is has been specified to avoid ambiguity. The results for antimicrobial resistance have been expanded. The number of Salmonella isolates that was subjected to AMR is small. The revised paragraphs are shown below:

On average, the prevalence of Salmonella spp. in beef and beef products was 1.25% (n = 5/ 400; Confidence interval (CI) = 0.4 - 3). The samples were mainly of gastrointestinal origin (3/400; 0.75%; CI = 0.2 - 2; Table 3), followed by the same number for beef kidney (n = 1/400; 0. 25%; CI = 0 - 1) and beef rib (n = 1/400; 0.25%; CI = 0 -1). From these 5 Salmonella positive samples, six isolates were obtained. The 6 isolates belong to 4 different serovars, namely S. Stanley (n = 3), S. Heidelberg (n = 1), S. Hadar (n = 1) and S. Enteritidis (n = 1) (Table 3).

Drug resistance was observed for S. Enteritidis only. The S. Enteritidis was resistant to tetracycline and the aminopenicillins, which are ampicillin and amoxicillin. The S. Stanley S. Heidelberg, and S. Hadar were susceptible to ampicillin, amoxicillin, chloramphenicol, cefotaxime, gentamicin, ciprofloxacin, kanamycin, tetracycline, cefoxitin, and trimethoprim – sulfamethoxazole.

Salmonella isolates tested positive to a minimum of 2 virulence genes and some serovars tested positive for up to 6 out of 8 virulence factors (Table 4). The virulence gene profile of each of the isolates was unique without repetition of the same combination of virulence genes (Table 4). Virulence gene agfA was predominant, and was observed in all of the 6 isolates. Both the invA and hilA genes were found in all except one isolate (Table 4). One of the three S. Stanley strains contained six of the 8 virulence genes, whilst S. Heidelberg harbored the least (2/8) number of virulence genes. The spvC gene was not detected in any of the 6 Salmonella isolates from this study.

Sincerely,

Prof. Evelyn Madoroba

(PhD Microbiology; MSc Biotechnology; Masters in Business Leadership; PGDip.Higher Education)

Round 2

Reviewer 3 Report

In this manuscript indeed the authors made some changes, that increased the quality of the paper (the results section), but no statistical analysis was mentioned again for both isolation of the Salmonella serovars and antimicrobial resistance, which in my opinion is not acceptable. Or if the authors consider statistical analyses are not needed, to explain why?

The author did not explain why in Table 3 in two samples of beef tripe S. Heidelberg and S. Hadar were detected, and in the case of beef intestines two S. Stainley were isolated, but in Table 4 are three samples of beef intestines and one of beef tripe. As I mentioned in the previous revision, this must be clarified! And also, the authors were recommended to add kidneys and intestines at Study design, Line 96!

And also, some minor recommendation was not considered: e.g. Line 391 - "From the 6 Salmonella isolates, 4 different serovars were identified. S. Stanley". The authors should change to "From the six Salmonella isolates, four different serovars were identified. S. Stanley". And there are also other examples like Lines 392, 395, 397, 398, 420, 423, 424, 436. 

Line 234: "Table 2" The authors should change to "Table 2". See the Instructions for authors! The same corrections should be made in the case of Line 300. 

Line 466: The year of publication should be bolded, as I recommended previously (see instructions for authors!!!).

The authors could see the recommendations in the attached file (v2). 

Author Response

Dear Editor,

We take this opportunity to thank Reviewer 3 for additional comments to enhance the quality of our manuscript. We have responded to the comments as follows:

Review 3 Comment 1: In this manuscript indeed the authors made some changes, that increased the quality of the paper (the results section), but no statistical analysis was mentioned again for both isolation of the Salmonella serovars and antimicrobial resistance, which in my opinion is not acceptable. Or if the authors consider statistical analyses are not needed, to explain why?

Response to comment 1: We have calculated the confidence intervals for the total prevalence and for the different food categories (line 261-263). We did not calculate differences between the different foods as the numbers are so small that it is apparent that there is no significant difference (this can also be deduced from the overlapping confidence intervals). Similarly, for the antimicrobial resistance and serovars, we did not calculate differences as the numbers were too small. For antimicrobial resistance, we found only one strain having resistance. We made also a statement on that in lines 407-407 of the discussion.

Reviewer 3 comment 2: The author did not explain why in Table 3 in two samples of beef tripe S. Heidelberg and S. Hadar were detected, and in the case of beef intestines two S. Stainley were isolated, but in Table 4 are three samples of beef intestines and one of beef tripe. As I mentioned in the previous revision, this must be clarified! And also, the authors were recommended to add kidneys and intestines at Study design, Line 96!

Response to comment 2: We included kidneys and intestines to the Study Design. During laboratory analyses for the presence of Salmonella, we recovered two Salmonella Stanley isolates from the same sample, we agree this is a duplicate sample that not should have been further analysed and should have been left out of the results. In order to avoid further ambiguity and confusion, we have removed the S. Stanley duplicate strain that was isolated from the same sample (from Tables 3 and 4), as well adapted the text where appropriate.

Reviewer 3 comment 3: And also, some minor recommendation was not considered: e.g. Line 391 - "From the 6 Salmonella isolates, 4 different serovars were identified. S. Stanley". The authors should change to "From the six Salmonella isolates, four different serovars were identified. S. Stanley". And there are also other examples like Lines 392, 395, 397, 398, 420, 423, 424, 436. 

Response to comment 3: We have adapted accordingly

Reviewer 3 comment 4: Line 234: "Table 2" The authors should change to "Table 2". See the Instructions for authors! The same corrections should be made in the case of Line 300. 

Response to comment 4: We have corrected accordingly

Reviewer 3 comment 5: Line 466: The year of publication should be bolded, as I recommended previously (see instructions for authors!!!).

Response to comment 5: We have adapted accordingly

Sincerely,

Evelyn Madoroba
